# Multilayered Polyurethane/Poly(vinyl alcohol) Nanofibrous Mats for Local Topotecan Delivery as a Potential Retinoblastoma Treatment

**DOI:** 10.3390/pharmaceutics15051398

**Published:** 2023-05-03

**Authors:** Radka Hobzova, Jakub Sirc, Kusum Shrestha, Barbora Mudrova, Zuzana Bosakova, Miroslav Slouf, Marcela Munzarova, Jan Hrabeta, Tereza Feglarova, Ana-Irina Cocarta

**Affiliations:** 1Institute of Macromolecular Chemistry, Czech Academy of Sciences, 162 06 Prague, Czech Republic; hobzova@imc.cas.cz (R.H.); sirc@imc.cas.cz (J.S.); kusumshrestha41@gmail.com (K.S.); slouf@imc.cas.cz (M.S.); 2Department of Analytical Chemistry, Faculty of Science, Charles University, 128 43 Prague, Czech Republic; barbora.mudrova@natur.cuni.cz (B.M.); zuzana.bosakova@natur.cuni.cz (Z.B.); 3Nanomedical Co., Ltd., 463 12 Liberec, Czech Republic; mmunzarova@nanomedical.cz; 4Department of Pediatric Hematology and Oncology, 2nd Faculty of Medicine, Charles University and Motol University Hospital, 150 06 Prague, Czech Republic; janhrabeta@gmail.com (J.H.);

**Keywords:** nanofibers, needleless electrospinning, multilayered structure, topotecan, prolonged release, retinoblastoma, cytotoxicity

## Abstract

Local chemotherapy using polymer drug delivery systems has the potential to treat some cancers, including intraocular retinoblastoma, which is difficult to treat with systemically delivered drugs. Well-designed carriers can provide the required drug concentration at the target site over a prolonged time, reduce the overall drug dose needed, and suppress severe side effects. Herein, nanofibrous carriers of the anticancer agent topotecan (TPT) with a multilayered structure composed of a TPT-loaded inner layer of poly(vinyl alcohol) (PVA) and outer covering layers of polyurethane (PUR) are proposed. Scanning electron microscopy showed homogeneous incorporation of TPT into the PVA nanofibers. HPLC-FLD proved the good loading efficiency of TPT (≥85%) with a content of the pharmacologically active lactone TPT of more than 97%. In vitro release experiments demonstrated that the PUR cover layers effectively reduced the initial burst release of hydrophilic TPT. In a 3-round experiment with human retinoblastoma cells (Y-79), TPT showed prolonged release from the sandwich-structured nanofibers compared with that from a PVA monolayer, with significantly enhanced cytotoxic effects as a result of an increase in the PUR layer thickness. The presented PUR-PVA/TPT-PUR nanofibers appear to be promising carriers of active TPT lactone that could be useful for local cancer therapy.

## 1. Introduction

Current research on the treatment of various oncological diseases has tended to focus on the local administration of active compounds to suppress the exposure of the organism to possible adverse effects. Therefore, polymeric drug delivery systems (DDSs), such as nanoparticles, liposomes, micelles, hydrogels, and nanofibers, have been intensively studied [1,2,3]. Electrospun nanofibers have recently emerged as a promising local delivery platform due to their unique morphological features, favorable drug loading capabilities, ease of use, and ability to be fabricated into various geometries with different sizes [4,5,6,7].

A fundamental aspect in the development of a suitable nanofibrous DDS is optimization of the design, composition, and morphology of the carrier in relation to the drug character and, above all, the setting of the drug release profile according to the therapeutic requirements. To control drug release kinetics, considering more advanced fibrous architectures, such as core–shell fibers, fibers filled with drug-loaded nanoparticles, or multilayered structures, is in the forefront of interest [8,9,10,11]. The challenge of prolonged drug release is most obvious in the case of hydrophilic drugs, as their incorporation into a simple nanofibrous monolayer mostly results in undesired rapid release (burst release) into the aqueous environment [12].

Needleless electrospinning (Nanospider™ technology, Elmarco Ltd., Liberec, Czech Republic) represents a universal method to produce micro/nanofibrous mats from various polymers [13,14] with high production capacity and therefore with great potential for use in clinical practice. Above all, this technique enables the incorporation of drugs by simple addition to the spinning polymer mixture, thereby providing nanofibrous drug carriers with high encapsulation efficiency and preserved pharmacological activity of the drug in vitro and in vivo [15,16,17]. Sequential needleless electrospinning can produce multilayered mats and hierarchically ordered nanofibrous structures for efficient drug delivery [18,19,20,21,22,23,24,25].

Reports concerning advanced nanofibrous structures as anticancer drug carriers have been summarized [6,7] to show their advantages in achieving therapeutically active drug concentrations in the target tumor cells along with a lower overall administered dose and less severe side effects. Particularly in the prevention of the local recurrence of breast [26], liver [27], lung [28] or bladder cancer [29], promising results have been demonstrated in in vivo animal models.

Topotecan (TPT) is a well-known anticancer agent commonly used for the treatment of various oncological diseases, including ovarian, small cell lung, cervical carcinoma [30], and intraocular retinoblastoma [31,32]. In most applications, TPT is used in a water-soluble hydrochloride, which is commercially available as Hycamtin. In solution, TPT occurs in two distinct forms—TPT carboxylate and TPT lactone, with the lactone form considered to be pharmacologically active. The ratio of the two forms in solution is pH dependent; at pH ≤ 4, the lactone is exclusively present, whereas the ring-opened hydroxy acid form predominates at physiological pH [33,34]. Therefore, when administered intravenously, TPT is delivered via a low pH infusion [30,35]. However, when circulating in blood, the active lactone form is partially converted to the carboxylate form at physiological pH before the drug reaches the desired site of action [36].

A suitable DDS for TPT should provide both stabilization of the TPT lactone form and its controlled release. In recent years, several TPT formulations have been studied, such as liposomes [37,38], lipid nanoparticles [39,40,41], injectable hydrogels [42,43,44], and polymer implants [45,46], showing sustained release and promising antitumor activity even in vivo in various tumor-bearing animal models [37,38,42,43]. However, to the best of our knowledge, there are no reports on DDSs for local TPT administration based on nanofibers, even though they may provide several advantages.

TPT, as a low molecular weight hydrophilic drug, would be desirable to incorporate into the hydrophilic nanofibers to ensure dissolution in the electrospun mixture and homogenous distribution within the polymer matrix. In this regard, poly(vinyl alcohol) (PVA), a biocompatible and nontoxic polymer which is used in biomedical applications [47,48] and for the preparation of nanofibrous materials [10], appears to be a suitable candidate. Some of the limitations of PVA fibers are their solubility and lower stability in aqueous media, but these can be overcome by crosslinking using glutaraldehyde vapors or thermal treatment. An indisputable advantage of PVA is that this polymer can be successfully electrospun from an acid solution, i.e., conditions under which the active lactone form of TPT can be maintained not only in solution but also in the dry nanofibrous carrier.

Previously, we successfully tested the concept of tri-layered nanofibrous mats composed of a middle PVA drug-loaded layer and PUR covering layers for the broad-spectrum antibiotic gentamicin [49]. Drug release experiments and in vitro testing of the antimicrobial activity proved the efficiency of the PUR cover layers to prolong the release of this low molecular weight drug.

The aim of this study was to prepare a nanofibrous drug carrier for local TPT delivery that could be useful in the treatment of retinoblastoma as an alternative to our previously reported bi-layered hydrogel carrier [46,50] and to be able to release TPT in its pharmacologically active lactone form. TPT was incorporated into hydrophilic PVA nanofibers. For sustained TPT release, hydrophobic PUR layers of different thicknesses were used to act as diffusion barriers for the drug. The proposed tri-layered sandwich-structured nanofibrous PUR-PVA.TPT-PUR mats were prepared via sequential needleless electrospinning technology (Nanospider™, Elmarco Ltd., Litvínov, Czech Republic). The effect of the PUR layer thickness on the kinetics of TPT release in vitro and on biological activity in human retinoblastoma cells in a multi-round experimental setup were investigated.

## 2. Materials and Methods

### 2.1. Materials

PVA (type Z 220, viscosity of 4 wt% in water solution at 20 °C 11.5–15.0 mPa·s, saponification degree 90.5–92.5 mol%) was supplied by Nippon Gohsei (Osaka, Japan). Polyurethane (PUR, Estane 5714 F1, viscosity of 15 wt% solution in THF 600–900 mPa·s) was obtained from Lubrizol Corp. (Wickliffe, OH, USA). TPT hydrochloride (≥98%) was obtained from U.S. Pharmacopeia (North Bethesda, MD, USA). *N*,*N*-Dimethylacetamide (Sigma-Aldrich, Prague, Czech Republic), toluene, phosphoric acid, and ammonium acetate were obtained in analytical grade (Penta, Prague, Czech Republic) and used as received. Water, acetonitrile, and methanol of HPLC grade were purchased from Merck (Prague, Czech Republic).

### 2.2. Preparation and Characterization of Tri-Layered Nanofibrous Mats

Nanofibrous mats were prepared by needleless electrospinning using Nanospider^TM^ technology (Nanovia Ltd., Litvinov, Czech Republic) [51]. The process parameters were individually optimized for each polymer. PVA was dissolved in a water/phosphoric acid mixture (90/10, *w*/*w*) at a polymer concentration of 11 wt%. The electrospinning conditions were as follows: distance between electrode 13 cm, voltage 45–55 kV/cm, relative humidity 25–30%, and temperature 22 °C. The area weight was 5 g/m^2^. PUR was dissolved in an *N*,*N*-dimethylacetamide/toluene mixture (2/1, *w*/*w*) at a polymer concentration of 10 wt%. The electrospinning conditions were as follows: distance between electrode 15 cm, voltage 5 kV/cm, relative humidity 25–30%, and temperature 22 °C. The area weight was 5 g/m^2^ or 15 g/m^2^. The TPT-loaded PVA layer was prepared by dissolving TPT in a PVA solution at a concentration of 1 wt% relative to the PVA mass and electrospinning under the same process parameters as those for the pure PVA mat.

The tri-layered sandwich structures, PUR-PVA-PUR, were prepared by sequential deposition of each layer. Nonwoven polypropylene material (ATEX, Milan, Italy) was used as a support for the deposition of the first PUR layer. Then, the PVA or TPT-loaded PVA layer was deposited, and finally, the top PUR layer was deposited. All layers were prepared using the same Nanospider^TM^ instrument, and only the process parameters were varied according to the type of polymer as described above. The area weights of the PUR layers in the sandwich-structured mats were 5 or 15 g/m^2^. The nanofibrous materials containing the PVA layer were thermally crosslinked in a drying oven at 120 °C for 10 min to ensure their stability in aqueous media. An overview of the prepared materials is given in Table 1.

The morphology of the nanofibrous mats was observed by scanning electron microscopy (SEM). The surface structure and fiber diameters of the samples cut from each individual monolayer were analyzed with a TS 5130 VEGA3 microscope (Tescan, Brno, Czech Republic) at an accelerating voltage of 15 kV after sputtering of the samples with a 4 nm thin platinum layer. The distributions of the fiber diameters were determined for single layers of PVA, PVA.TPT, and PUR(15) by measuring the diameter of 10 randomly selected fibers from each of three SEM micrographs at a magnification of 10,000× using ImageJ software (*n* = 30). The fiber distribution was calculated as the frequency of the fiber diameters and is expressed as a percentage. The cross-sections of the tri-layered mats were visualized by means of an MAIA3 SEM microscope (Tescan, Brno, Czech Republic). The cross-sections were prepared by fracturing in liquid nitrogen: a small stripe of the sample was submerged in liquid nitrogen for 5 min and then fractured using two pairs of tweezers. The fracture surface was sputtered with a thin (4 nm) platinum layer, fixed to an aluminum support with double-sided adhesive carbon tape and observed with a SEM microscope using a chamber secondary electron detector at an accelerating voltage of 3 kV.

Stability of PVA, PUR monolayers, and PUR/PVA sandwiches in water was determined as follows: the nanofibers were cut into pieces of 2 × 5 cm^2^ and weighing approximately 10 to 30 mg. The cut samples were dried under vacuum at room temperature, weighed, placed in sealed bottles with 10 mL of water, and then left at 37 °C in an incubation shaker (IKA KS 4000i, 120 rpm, Fisher Scientific, Prague, Czech Republic). At a particular time period, three specimens were filtered and dried under vacuum at room temperature to a constant weight. Stability of the materials expressed as weight loss (WL, %) was calculated according to Equation (1):*WL* (%) = (*w*_0_ − *w*_t_)/*w*_0_ × 100(1)
where *w*_0_ is the initial weight of the dried sample and *w*_t_ is the weight of the dried sample after a particular time of contact with aqueous medium. *WL* is expressed as mean ± standard deviation (*n* = 3).

### 2.3. Drug Loading and In Vitro Drug Release

#### 2.3.1. Determination of the TPT Form (Lactone/Carboxylate) and Total TPT Content in the Nanofibrous Mats

The TPT content in the nanofibrous mats was determined by extraction experiments as follows. Round targets (discs with a diameter of 8 mm and weight of approximately 0.25 mg) were cut from TPT-loaded mats using a biopsy punch (Stiefel from Servoprax, Wesel, Germany), weighed, and separately immersed in 2 mL of methanol for 24 h at 4 °C with gentle shaking.

The concentration of the released TPT was detected with an HPLC-FLD apparatus with a Shimadzu LC-20AD HPLC system (Shimadzu, Kyoto, Japan) to detect for the TPT lactone and carboxylate forms separately.

Separation was performed on an Ascentis Express C18 column (3 mm × 150 mm, 5 µm particle size) (Merck, Prague, Czech Republic) with a linear gradient elution. The mobile phase consisted of 5 mM acetate ammonium buffer pH 4.5 (A) and methanol (B). The gradient elution program was as follows: 30% B (0 min), 30–70% B (10 min), 70% B (1 min), 70–30% B (0.1 min) and 30% B (4.9 min) with a total run time of 16 min. The mobile phase flow rate was 0.5 mL/min. The sample injection volume was 10 µL. The excitation wavelength for fluorescence detection of both TPT forms was set to 361 nm, and the emission wavelength was set to 527 nm.

A stock solution of TPT (1 mg/mL) was prepared in deionized water and immediately diluted with 5 mM ammonium acetate buffer adjusted to pH 3 to prepare two standard solutions (TPT concentrations of 1 µg/mL and 100 ng/mL) for TPT lactone calibration. This step was repeated with 5 mM ammonium acetate at pH 10 for TPT carboxylate calibration. The pH 10 stock solution was kept at 8 °C for 3 h after its preparation to reach equilibrium. Two series of calibration standards for the TPT lactone and TPT carboxylate were prepared by diluting the 1 µg/mL and 100 ng/mL standard solutions at the appropriate pH, i.e., each contained 10% (*v*/*v*) 5 mM ammonium acetate buffer (pH 3 or 10), 50% (*v*/*v*) ice-cold methanol, and 40% (*v*/*v*) deionized cold water (8 °C). The total volume of each calibration standard was 200 µL. Eight calibration standards were prepared at concentrations of 0.5, 1, 5, 10, 25, 50, 75, and 100 ng/mL for both TPT lactone and TPT carboxylate and analyzed twice using the HPLC-FLD method. All calibration curves were linear in the whole concentration range, with *R*^2^ values over 0.99 and a relative standard deviation below 5%.

Determination of the TPT lactone, TPT carboxylate, and total TPT contents in the nanofibrous mats was performed with two independent measurements, with each experiment performed in triplicate; the results are expressed as the average value of micrograms of TPT released from an 8 mm disc ± standard deviation (SD, *n* = 6). The drug loading efficiency (LE, %) was calculated using Equation (2):LE(%) = *m*_real_/*m*_theor_ × 100(2)
where *m*_real_ and *m*_theor_ are the experimentally determined and the theoretically calculated TPT content in the nanofibrous mats. The theoretical TPT amount per 8 mm disc was calculated from the area weight of the PVA mat (5 g/m^2^), considering that the addition of TPT to the spun mixture was 1 wt% relative to the PVA mass.

#### 2.3.2. Determination of TPT Release Kinetics Using HPLC-FLD

To determine the TPT release kinetics, TPT quantification was performed according to the HPLC-FLD method described earlier [46]. Briefly, a Shimadzu LC-20AD HPLC system (Shimadzu, Kyoto, Japan) fitted with a Chromolith RP-18e column (Merck, Prague, Czech Republic) was used under the following conditions: mobile phase ACN/water 70/30 *v*/*v*, flow rate 1.5 mL/min, injection volume 10 µL, and fluorescence detection at an excitation wavelength at 361 nm and an emission wavelength at 527 nm.

Round targets (8 mm discs) were cut from TPT-loaded mats, incubated in 2 mL of water (pH 6.7) and kept in the dark at 4 °C. At predetermined time intervals (5, 10, 15, 20, 25, 30, 45 min, 1, 1.5, 2, 3, 4, and 5 h), 2 mL of water was withdrawn and analyzed by HPLC-FLD. Then, fresh water with the same volume was added for replacement. The experiments were carried out in triplicate. The cumulative drug release (CR, %) was calculated using Equation (3):*CR* (%) = {[(*c*_n_ + ∑*c*_n−1_) × V]/*m*_theor_} × 100,(3)
where *c*_n_ and *c*_n−1_ are the concentrations of drug (µg/mL) in the release medium after *n* and *n* − 1 withdrawing steps, respectively; *n* is the number of withdrawing steps; and *V* is the volume of release medium. The resulting CR is expressed as the average value ± SD.

### 2.4. In Vitro Cytotoxicity of the TPT-Loaded Nanofibrous Mats

The in vitro cytotoxicity of the TPT-loaded nanofibrous mats (PVA.TPT, PUR(5)-PVA.TPT-PUR(5), and PUR(15)-PVA.TPT-PUR(15)) was evaluated using the human retinoblastoma cell line Y-79 (HTB-18, ATCC, VA, USA). Discs 4 mm in diameter were cut from nanofibrous mats with a sterile biopsy punch. Cells were cultured in RPMI 1640 medium (Sigma-Aldrich, Prague, Czech Republic) supplemented with 15% bovine serum and 1% penicillin—streptomycin in a humidified atmosphere at 37 °C and 5% CO^2^. To see the effect of the sandwich structure on prolonged drug release, the experiment was set up in 3 rounds (Figure 1). In the 1st round, nanofibrous samples were placed on 24-well permeable transwell inserts with 400 μL of culture medium (Corning, NY, USA), added to pre-incubated cells (24 h, 600 μL of media, cell density 1 × 10^5^) in a 24-well plate and incubated for 4 or 24 h. Then, the transwell insert with the nanofibrous samples was transferred to a new 24-well plate with fresh cell culture and incubated for 24 h to undergo the 2nd round. In the 3rd round, the samples were incubated for another 24 h with a new fresh cell culture. Cell viability after 48 h in each round was determined by PrestoBlue assay (ThermoFisher Scientific, Waltham, MA, USA) and was calculated as a percentage of the control cells at each time point. The whole procedure was repeated at least twice.

Data are shown as the average ± SD. Statistical analysis was performed by ANOVA/post hoc Tukey’s HSD test. A value of *p* < 0.05 was considered significant.

## 3. Results and Discussion

### 3.1. Characterization of the Nanofibrous Mats

The nanofibrous structure of the prepared monolayered mats observed by SEM exhibited a randomly oriented, bead-free structure with a smooth surface (Figure 2, left side images). The incorporation of TPT into the PVA nanofibers had no obvious effect on the morphology. In Figure 2b, no crystals or aggregates of the drug are visible on the fiber surface or outside the fibers, indicating that the drug is compatible with the polymer solution, leading to the incorporation of TPT inside the fibers and thus a homogenous distribution within the PVA matrix. This is essential to prevent drug deposition on the fiber surface and the consequent undesired initial burst release [52].

The fiber diameter distribution curves are presented in Figure 2 (right side images). The most frequent diameters for PVA and PVA.TPT were in the range of 100 to 300 nm. With the incorporation of TPT, the mean fiber diameter increased slightly, from 168 ± 81 nm for PVA to 188 ± 57 nm for PVA.TPT. PUR exhibited a broader distribution, with the most frequent fiber diameters ranging from 200 to 500 nm and a mean diameter of 355 ± 138 nm.

The structure of the tri-layered PUR-PVA-PUR nanofibrous mats was also evaluated by SEM. Figure 3 illustrates the cross-sectional micrographs of the sandwiches with an apparently thicker layer of PUR fibers in PUR(15)-PVA-PUR(15). Most of the SEM micrographs revealed tightly bonded layers of PVA and PUR fibers, while in some micrographs, a certain gap between the layers appeared. However, this gap may form during sample preparation for SEM imaging.

The SEM observations were in accordance with our previous research, which showed that the direct incorporation of additional substances by simple addition to the electrospun mixture can provide nanofibrous drug-loaded carriers that do not cause a substantial impact on their fibrous structure [15,17,49]. Generally, an appropriate composition of the electrospun mixture with sufficient solubility of both the polymer and the drug given by their hydrophilic/hydrophobic character results in drug carriers with drug molecules homogenously distributed within the fiber matrix [52].

Stability of polymeric nanofibrous matrices was investigated under conditions consistent with in vitro drug release experiments (i.e., 37 °C, water, 72 h). Figure 4 represents the results of the weight loss (WL) as a function of time, showing that WL does not change over the time and reaches values of approximately 12–15%, 5%, and 2.5% for PVA, PUR(5)-PVA-PUR(5), and PUR(15)-PVA-PUR(15), respectively. In the case of PVA, WL occurs already in the first time period and can therefore be attributed to the dissolution of the polymer chains that have not been crosslinked during the thermal treatment applied. The PUR mats were stable throughout, showing WL of less than 2% (data not presented). The WL in the case of sandwiches is given by loss of PVA which proportionally corresponds to the mass of PVA in each sandwich.

Although thermal crosslinking did not ensure complete insolubility of PVA, it is a suitable crosslinking method that can be used for the preparation of drug-loaded nanofibers. Importantly, the polymer matrices crosslinked in this way were not toxic to cells themselves and retained the cytotoxic activity of the incorporated drug (see results below).

### 3.2. Drug Loading

Due to its relatively low therapeutically effective concentration (the reported IC50 value for the Y-79 cell line is approximately 30 nM [53,54,55], and the recommended dose for systemic administration is 0.75–2.3 mg/m^2^ body surface area according to the protocol [30]), TPT was added to the electrospun mixture at 1 wt% relative to the PVA polymer. The parameters of the needleless electrospinning process were set to achieve an area weight of the PVA nanofibrous layer approximately 5 g/m^2^, providing mats convenient for manipulation and with a sufficient amount of incorporated drug to measure the release profile and biological activity. Similar conditions were also followed for the tri-layered mats to keep the PVA thickness constant. It is worth noting that the dosage of TPT can be easily adjusted according to the required therapeutic effect by increasing the area weight of the mat or by increasing the amount of the drug added to the spun mixture.

The real TPT contents (µg/disc) in the TPT-loaded mats were determined by immersing 8 mm discs into methanol and subsequently quantifying TPT by HPLC. The theoretical TPT contents calculated from the disc surface area compared with the experimentally determined contents are shown in Table 2.

Loading efficiencies of 86%, 90%, and 108% for PVA.TPT, PUR(5)-PVA.TPT-PUR(5), and PUR(15)-PVA.TPT-PUR(15), respectively, indicate that the set conditions for electrospinning the PVA solution with TPT were appropriate for complete dissolution of PVA and TPT and their subsequent transfer from solution to homogeneous nanofibers, with a composition that is in good agreement with the initial electrospun mixture.

Methanol was chosen as the release agent for TPT because it can preserve the lactone and carboxylate forms [56]. HPLC analysis revealed that the PVA nanofibrous mats contained 97.8 to 100% TPT in the biologically active lactone form. Apparently, the acidic solvent composed of 10% phosphoric acid used in the electrospun mixture converts all TPT to the lactone form, which is subsequently retained in the PVA nanofibers even after solvent evaporation. Thus, when used in vivo, the released TPT may produce pharmacological activity. This is the main advantage of PVA nanofibrous carriers over other drug delivery systems. For example, in hydrogels, TPT loading proceeds at a neutral pH in which the inactive carboxylate form of TPT prevails, and therefore, the final hydrogel drug carrier may not contain only the TPT lactone.

### 3.3. In Vitro TPT Release Study

In vitro release experiments play an important role in the systematic development of drug delivery systems; however, setting the release conditions and experimental design always represent a certain compromise between trying to mimic in vivo conditions as closely as possible and taking into account the properties of the drug, such as its solubility and stability in aqueous solutions. Previously, we reported the results of TPT stability in solution as a function of temperature, pH, type of release medium, and drug concentration [46]. The highest TPT stability was found in water as the release medium at 4 °C in the dark. Therefore, we used these conditions in this study. Although such conditions do not exactly resemble the in vivo environment, as they do not correspond to physiological temperature, pH, or ion strength, we assume that these play a less important role compared with the negative effect of low TPT stability at 37 °C in PBS or plasma. Extensive TPT decomposition during the release experiment would not enable a reveal of the effect of PUR layer coverings on the TPT release profiles.

The obtained TPT release profiles are shown in Figure 5. The TPT-loaded PVA monolayered mats without PUR coating released almost all of the incorporated TPT within approximately 30 min. Overlaying such a hydrophilic mat with a thinner (5 g/m^2^) layer of PUR nanofibers prolonged the TPT release by a factor of five to approximately 2.5 h. A thicker sandwich structure with 15 g/m^2^ PUR layers released one-third of the TPT content within 2.5 h, followed by further release for an additional 2.5 h. Longer time periods were not evaluated, as TPT decomposition can negatively influence the concentration observed; however, further TPT delivery can be expected, as long-term biological activity of released TPT was observed in in vitro cell experiments (see below).

TPT, which is soluble in water, is released from the nanofibrous mats mainly by a diffusion-driven mechanism. In the monolayered PVA system, TPT is easily accessible to the aqueous medium inside the hydrophilic fibrous mats, resulting in its rapid dissolution and subsequent release. It has been demonstrated that PUR coating significantly slows TPT release, with this effect being more pronounced with thicker PUR layers. This effect is due to the longer distance from the middle PVA reservoir to the sandwich periphery through which TPT must diffuse, as well as through the hydrophobic layers. Furthermore, the effect of delayed release with increasing thickness of the PUR layers could be caused by the higher fiber density (lower porosity) produced by the longer time of fiber deposition during electrospinning and the consequent greater compression of the nanofibrous structure, as we described in our previous work [13]. In summary, release experiments confirmed that the thickness of the PUR cover layer can be efficiently used to decrease the burst release of TPT in the initial stage and prolong its release.

It is important to note that the behavior of DDSs under real in vivo conditions is likely to be different due to factors such as higher temperature, ion concentration, or the presence of peptides and proteins. Although, the nanofibrous mats tested did not reveal any disintegration within the release experiments (no degradation was evident), under physiological conditions in vivo, enzymatic activity or mechanical stress may have an impact on degradation of the polymer matrix.

### 3.4. In Vitro Cytotoxicity

The in vitro cytotoxicity of TPT-loaded nanofibrous mats was evaluated using the human retinoblastoma cell line Y-79 with monolayered PVA.TPT and tri-layered PUR-PVA.TPT-PUR sandwich mats. We also tested the toxicity of TPT-free sandwich nanofibrous mats, i.e., PUR(5)-PVA-PUR(5), which served as blank samples, to the cells of this line.

Often, in vitro activity is monitored via a simple experiment by immersing the drug-loaded sample in the cell solutions and determining the cytotoxic effect of the drug after a given period of time [19,22]. However, the final cytotoxic effect is mainly produced by the burst release of the drug in the initial time period. Our TPT-loaded nanofibrous mats are intended for prolonged drug release, and therefore, a multiple-round experiment was considered. We designed our biological study as a 3-round experiment where, in the 1st round, two times of pre-incubation (2 and 24 h) were applied in order to eliminate the effect of the drug burst release on cell viability. The next rounds were performed to determine the long-term efficacy of cytotoxicity where, in each round, the same samples were used and tested with fresh, untreated cells. Previously, we had successfully used this experimental setup to test the cytotoxicity of paclitaxel released from poly(lactide)/poly(ethylene glycol) nanofibrous carriers [16].

Figure 6a shows that the nanofibrous mats without TPT were not toxic to Y-79 cells, and the proportion of viable cells was above 90% in the 3-round experiment at all time periods tested. Although this mat was not tested in cell lines other than Y-79, these results predict the nontoxicity of polymeric nanofibrous mats composed of PVA and PUR.

The cytotoxic effect of the TPT released from the PVA monolayer and tri-layered mats within the 3-round experiment is shown in Figure 6b–e. In the case of the TPT-loaded PVA monolayer (without PUR cover layers), approximately 45% cell viability was observed during the 1st round, followed by viabilities between 80 and 90% in rounds 2 and 3 (Figure 6b). This suggests that the majority of the TPT was released during the 1st round, mainly in the first few hours, as no significant difference was found between 4 and 24 h of incubation. The rapid release of most of the TPT from the PVA mat without PUR is consistent with the in vitro experiments (see above) with significantly lower cytotoxicity in rounds 2 and 3, when viability approached that of the blank control samples.

The results of the cytotoxicity assessments obtained for the TPT-loaded sandwich mats are shown in Figure 6c,d. It can be seen that overlaying the TPT-loaded PVA layer with PUR layers had a significant effect on reducing the viability of retinoblastoma cells and that this effect persisted even in rounds 2 and 3 in contrast to the PVA.TPT monolayer. In the case of the thicker PUR(15)-PVA.TPT-PUR(15) sandwich, the cell viability in the 3rd round was still approximately 60%. Both sandwich mats exhibited a significant reduction in cytotoxic effects between rounds 1 and 2, with viabilities increasing from approximately 40% to 70% for the thinner sandwich and from 35% to 60% for the thicker one. The differences in viability in rounds 2 and 3 were not significant. This suggests that a significant portion of the TPT was released during the 1st round, followed by the release of smaller portions of the drug in subsequent rounds.

Figure 6e compares the viabilities determined in each round (with 24 h of incubation in the 1st round) for all three mats tested. There were no significant differences in round 1. Apparently, the initial burst release of the TPT had sufficient biological effect to decrease the cell viability to approximately 40%, even for the PUR-covered mats. In rounds 2 and 3, the differences between particular nanofibrous mats are significant. Thus, it can be concluded that it is possible to modulate the TPT release profile by controlling the thickness of the electrospun cover layers over the drug-loaded layer.

In vitro biological experiments on cell lines proved prolonged biological activity and confirmed the premise of using PUR-PVA-PUR constructs as a TPT delivery system. Compared with the hydrogel system for periocular delivery presented in our previous work [46,50,57], this system has several advantages: (i) TPT is preserved in its active lactone form; (ii) TPT release can be controlled by inactive PUR covering layers; (iii) TPT is incorporated into the DDS during the implant production, allowing for easier implant preparation prior to in vivo application. Compared with another DDS system in the form of compressed polycaprolactone tablets [45], nanofibrous mats provide a convenient and versatile design feasible for periocular implantation with a tight fit to the eyeball. Moreover, the active drug-releasing layer can be easily covered with an impermeable polymeric shell (e.g., made from poly(2-ethoxyethyl methacrylate [46]) that protects the surrounding sensitive tissue from cytotoxicity and directs drug release toward the eyeball. In vivo experiments with such a proposed implant will be the subject of future work.

## 4. Conclusions

The anticancer agent TPT was successfully incorporated into a PVA nanofibrous layer by a needleless electrospinning technique due to the hydrophilic character of both components, resulting in a homogenous dispersion of the drug in the polymer matrix with a loading efficiency of more than 85%. Sequential electrospinning enabled the preparation of sandwich-structured nanofibrous mats with TPT-loaded PVA mats as the inner layer overlaid by hydrophobic PUR cover layers of various thicknesses. SEM measurements proved that the incorporation of TPT or the layering of PUR and PVA mats had no impact on the nanofibrous structure compared with simple monolayers.

The HPLC-FLD method was optimized to determine the contents of the lactone and carboxylate forms of TPT. It was proven that the set conditions for TPT-loaded PVA mat preparation ensure the incorporation of the therapeutically active lactone form into the polymer matrix, as more than 97% TPT lactone was determined to be in the methanol extracts from the nanofibers.

Both in vitro release experiments and cell experiments showed that the sandwich-structured mats successfully slowed the release of hydrophilic TPT into the surrounding environment, thereby prolonging its duration of action. The cytotoxic effect in the human retinoblastoma cell line Y-79 in the 3-round experiment was significantly higher for the sandwiches compared with the PVA.TPT monolayer, and this effect was enhanced by increasing the thickness of the PUR layers.

Our results demonstrate that it is possible to modulate the TPT release profile by controlling the thickness of hydrophobic cover layers over the drug-loaded layer. Thus, our presented PUR-PVA.TPT-PUR sandwich nanofibers with sustained drug release promise to provide TPT in its biologically active form for a prolonged period of time. Moreover, a certain advantage of the PVA/PUR multilayered nanofibrous carrier of TPT is its relatively easy dosing with the active agent, since the portion of TPT in PVA fibers can be increased (from 1% to at least 10%) or the thickness of the PVA mat in the sandwich can be increased (from 5 g/m^2^ to at least 25 g/m^2^). The combination of modulating the TPT concentration in the loaded layer and its thickness represents an efficient way to optimize the amount of TPT released according to the therapeutic requirements and thus a potential for use in local tumor therapy.

In clinical practice, in addition to retinoblastoma, TPT is used to treat small cell lung carcinoma, ovarian carcinoma, and cervical carcinoma. In particular, in the treatment of cervical carcinoma, due to its location, local TPT application via a nanofibrous carrier could also be beneficial.

## Figures and Tables

**Figure 1 pharmaceutics-15-01398-f001:**
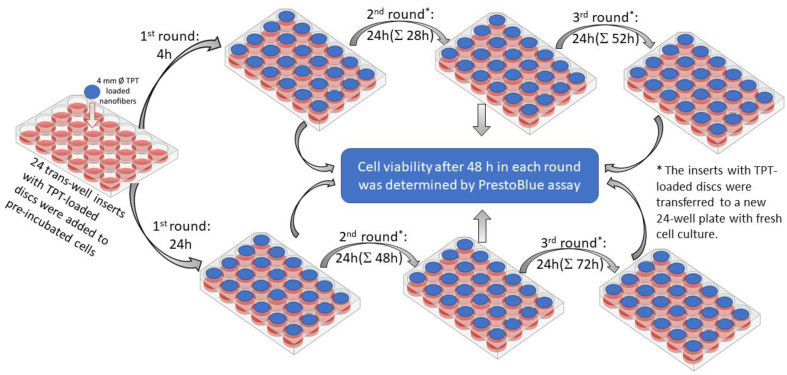
Scheme of the 3-round experiment.

**Figure 2 pharmaceutics-15-01398-f002:**
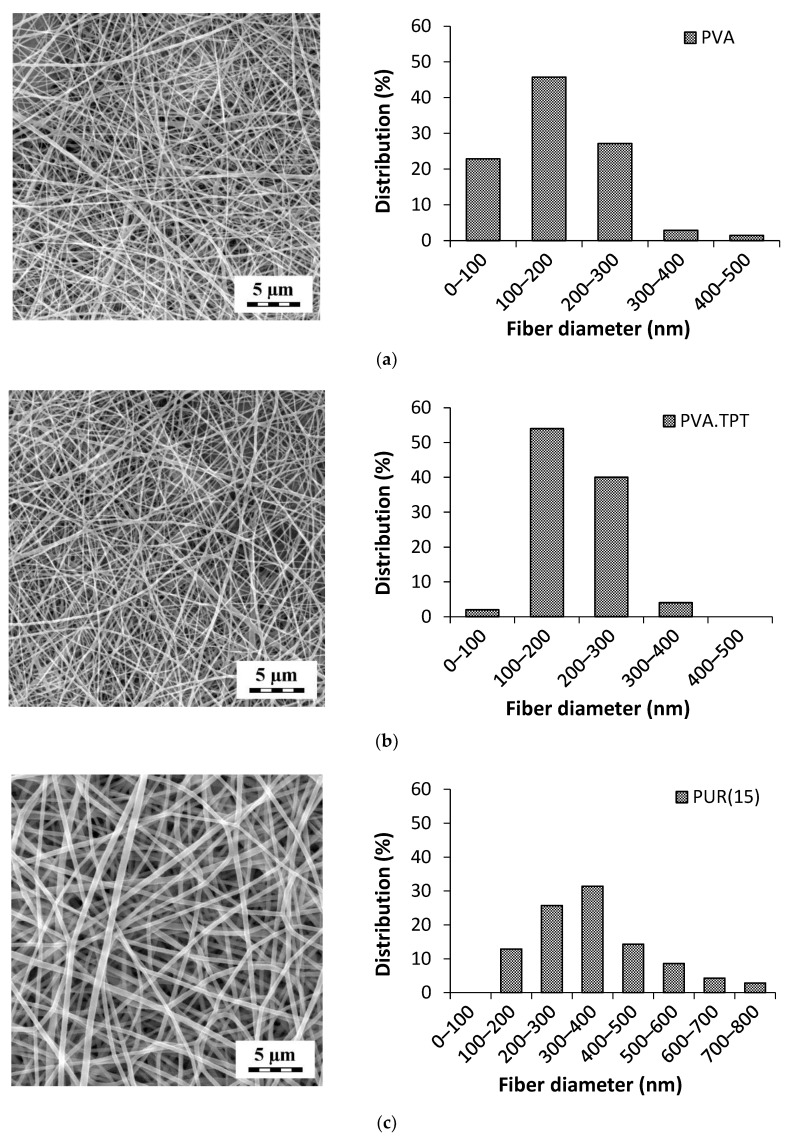
SEM micrographs (left) and the fiber diameter distribution of the PVA and PUR monolayers: (**a**) PVA mat; (**b**) PVA.TPT mat; and (**c**) PUR(15) mat. Magnification 10,000×.

**Figure 3 pharmaceutics-15-01398-f003:**
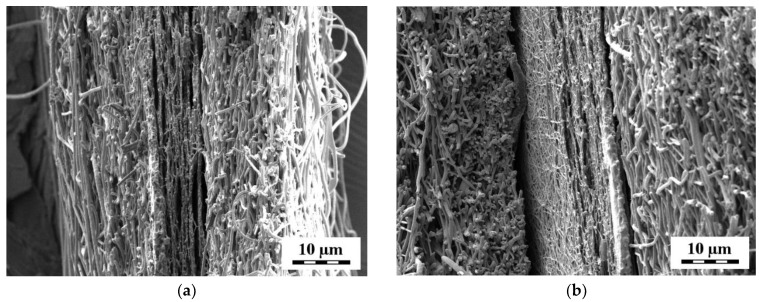
Cross-sectional SEM micrographs of tri-layered nanofibrous mats: (**a**) PUR(5)-PVA-PUR(5) and (**b**) PUR(15)-PVA-PUR(15). Magnification 3300×.

**Figure 4 pharmaceutics-15-01398-f004:**
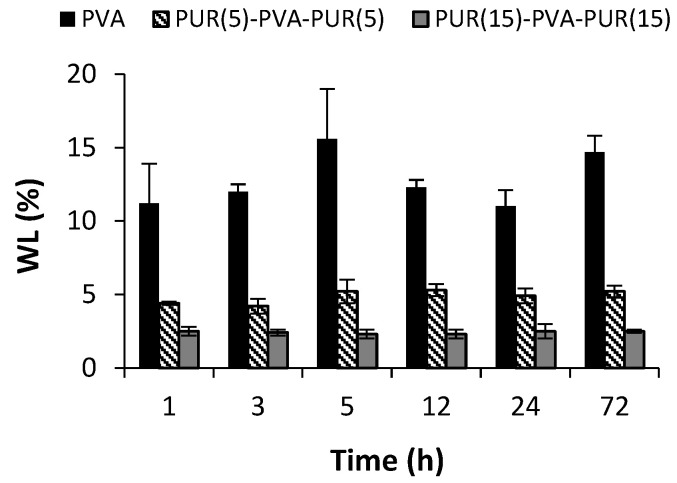
Weight loss (WL) of nanofibrous mats as a function of immersion time in water at 37 °C under shaking. The WL was calculated according to Equation (1). Error bars were calculated using standard deviation (*n* = 3).

**Figure 5 pharmaceutics-15-01398-f005:**
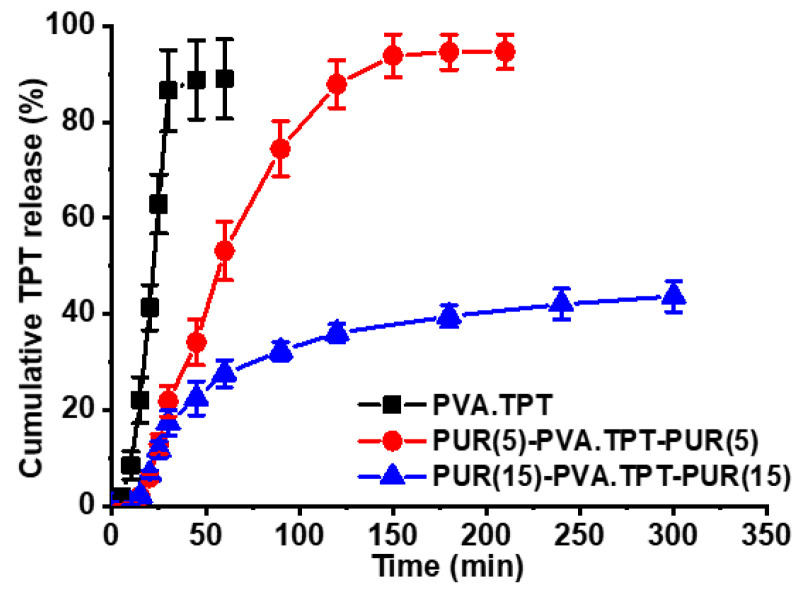
TPT release profiles from PVA.TPT, PUR(5)-PVA.TPT-PUR(5), and PUR(15)-PVA.TPT-PUR(15) nanofibrous mats (water, pH 6.7, 4 °C). The results were determined by HPLC. Data are expressed as the average values. Error bars were determined from the standard deviation (*n* = 3).

**Figure 6 pharmaceutics-15-01398-f006:**
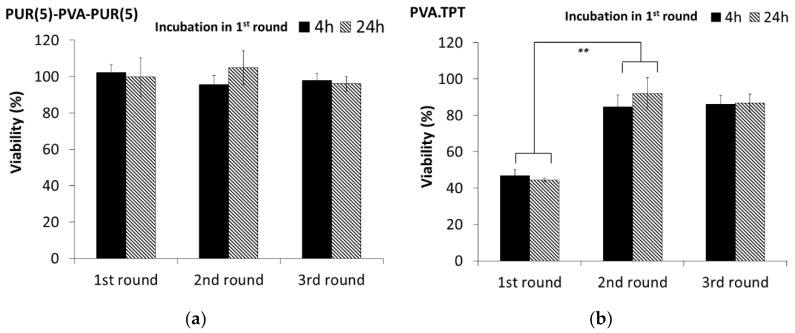
Viability of human retinoblastoma Y-79 cells in the 3-round experiment after treatment with nanofibrous mats: (**a**) PUR(5)-PVA; (**b**) PVA.TPT; (**c**) PUR(5)-PVA.TPT-PUR(5); (**d**) PUR(15)-PVA.TPT-PUR(15); and (**e**) comparison of all mats tested. The values connected by the drawn lines represent significant differences between the incubation rounds or between particular nanofibrous mats within the same incubation round (* *p* < 0.05, ** *p* < 0.01).

**Table 1 pharmaceutics-15-01398-t001:** Overview of the prepared nanofibrous mats.

Sample Code	Area Weight (g/m^2^)		Fiber Diameter ^1^ (nm)
	PVA	PUR	
PVA	5	-	168 ± 80
PVA.TPT	5	-	188 ± 56
PUR(15)	-	15	355 ± 137
PUR(5)-PVA-PUR(5)	5	5	
PUR(15)-PVA-PUR(15)	5	15	
PUR(5)-PVA.TPT-PUR(5)	5	5	
PUR(15)-PVA.TPT-PUR(15)	5	15	

^1^ Calculated as the mean value ± standard deviation (*n* = 30).

**Table 2 pharmaceutics-15-01398-t002:** TPT contents in the TPT-loaded nanofibrous mats and drug loading efficiency.

Sample	Release Medium/Time (h)	Content of TPT		LE ^3^ (%)	Percentage of TPT Lactone Form ^4^ (%)
		Theoretical ^1^ (µg/disc)	Real ^2^ (µg/disc)		
PVA.TPT	Methanol/24 h	2.51	2.15 ± 0.23	85.7	100 0
PUR(5)-PVA.TPT-PUR(5)			2.27 ± 0.16	90.4	97.8
PUR(15-PVA.TPT-PUR(15)			2.72 ± 0.41	108.4	98.7

^1^ Calculated from the area of an 8 mm disc with a theoretical area weight of 5 g/m^2^. ^2^ Total TPT content determined by HPLC; average value ± SD (*n* = 6). ^3^ Loading efficiency (percentage) determined as the ratio of the real and theoretical TPT contents. ^4^ Content of TPT lactone from the total TPT content determined by HPLC.

## Data Availability

Data are available upon request.

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
