# Peer review of "Multilayered Polyurethane/Poly(vinyl alcohol) Nanofibrous Mats for Local Topotecan Delivery as a Potential Retinoblastoma Treatment"

_pharmaceutics, 2023, doi:10.3390/pharmaceutics15051398_

Round 1
Reviewer 1 Report
This manuscript by Hobzova et al describes preparation of topotecan (TPT)-loaded three-layered nanofibrous mat using poly(vinyl alcohol) (PVA) and polyurethane (PUR) by sequential electrospinning technique. Further, it investigates the physicochemical characteristics, TPT release kinetics and in vitro cytotoxicity of the prepared nanofibrous mat. The authors claim that this nanofibrous mat could be useful for local TPT delivery in the treatment of retinoblastoma. However, no in vivo experiments were considered to validate this main claim. Therefore, the authors should evaluate the potential of the TPT-loaded nanofibrous mat using an appropriate in vivo model. The detailed comments are given below.
Comments
1. Great emphasis has been placed on the use of TPT-loaded PUR/PVA nanofibrous mats for retinoblastoma treatment. Further, the authors are concluded these nanofibrous mat could be useful in treatment of small cell lung carcinoma, ovarian carcinoma and cervical carcinoma. However, no in vitro/in vivo experiments are performed/presented to substantiate the potential of the nanofibrous mat for these applications.
2. A major concern is no detailed in vivo experiments are performed to demonstrate the potential of the TPT-loaded nanofibrous mats for retinoblastoma treatment, other than evaluating in vitro cytotoxicity.
3. It is difficult to comprehend the rationale behind 3-round cytotoxicity experiments? Please discuss these details in the manuscript.
4. In vitro drug release study (Figure 4a) shows that TPT drug loaded into both uncoated and PUR-coated nanofibrous mat were almost released within 5 h. Further, it also indicates that increase in the thickness of the hydrophobic PUR layer decrease the TPT release from the fibrous mat. However, in vitro cytotoxicity study (Figure 5e) indicates no significant difference in viabilities between uncoated and PUR-coated mats in the 1st round. Why is this discrepancy?
5. For biomedical applications, the stability of electrospun nanofibrous mats in physiological conditions is important and needs to be characterized. How about the stability of the PUR/PVA nanofibrous mat under physiological condition?
6. How is TPT-loading content in the nanofibrous mat calculated? Please provide the details for calculating the TPT loading efficiency (LE), similar to that of cumulative drug release (equation 1).
7. Please expand the acronym PUR/PVA in the title.
8. What is the pH of water used for drug release study?
Author Response
Reviewer 1:
This manuscript by Hobzova et al describes preparation of topotecan (TPT)-loaded three-layered nanofibrous mat using poly(vinyl alcohol) (PVA) and polyurethane (PUR) by sequential electrospinning technique. Further, it investigates the physicochemical characteristics, TPT release kinetics and in vitro cytotoxicity of the prepared nanofibrous mat. The authors claim that this nanofibrous mat could be useful for local TPT delivery in the treatment of retinoblastoma. However, no in vivo experiments were considered to validate this main claim. Therefore, the authors should evaluate the potential of the TPT-loaded nanofibrous mat using an appropriate in vivo model. The detailed comments are given below.
Comments
- Great emphasis has been placed on the use of TPT-loaded PUR/PVA nanofibrous mats for retinoblastoma treatment. Further, the authors are concluded these nanofibrous mat could be useful in treatment of small cell lung carcinoma, ovarian carcinoma and cervical carcinoma. However, no in vitro/in vivo experiments are performed/presented to substantiate the potential of the nanofibrous mat for these applications.
We agree with the Reviewer´s comment, that to substantiate the usefulness of the nanofibrous mats in the treatment of various cancers more experiments would be needed. However, this was not the aim of this work. For this special issue dealing with eye drug delivery, we have focused on the retinoblastoma therapy. Based on the discussion with our colleagues - oncologists, we have just mentioned at the end of “Conclusion” the perspectives of these materials for other potential applications, as a topic for the further studies.
- A major concern is no detailed in vivo experiments are performed to demonstrate the potential of the TPT-loaded nanofibrous mats for retinoblastoma treatment, other than evaluating in vitro cytotoxicity.
In vivo experiments certainly play a crucial role in the systematic development of the drug delivery systems. Previously, we have reported on our results regarding the development of a hydrogel implant for TPT delivery into the eyeball, where the preparation and characterization of the material, its in vitro release and biological activity were first published and then in vivo testing was published separately (Cocarta 2019, Hobzova 2021, Kodetova 2022). Also, in this case, the in vivo experiments, including design of the implant, ways of the implant administration into the animal model, testing of pharmacokinetics and treatment efficiency, will be a content of one or more further research papers.
- It is difficult to comprehend the rationale behind 3-round cytotoxicity experiments? Please discuss these details in the manuscript.
We agree with the Reviewer´s comment that the reason for the 3-round experiment is not clear from the manuscript. We have made changes to the text to explain the purpose of such designed biological experiment, lines 375-384.
- In vitro drug release study (Figure 4a) shows that TPT drug loaded into both uncoated and PUR-coated nanofibrous mat were almost released within 5 h. Further, it also indicates that increase in the thickness of the hydrophobic PUR layer decrease the TPT release from the fibrous mat. However, in vitro cytotoxicity study (Figure 5e) indicates no significant difference in viabilities between uncoated and PUR-coated mats in the 1st round. Why is this discrepancy?
In accordance with our previous experience, the in vitro biological activity of polymeric drug release devices does not exactly correspond to the drug release kinetics determined in excess of the release medium, in addition under shaking. In this case, there is no discrepancy, but the results are relatively consistent. All types of nanofibrous materials, both single and multi-layer, release a large portion of TPT quite fast (within 5h or less), causing apparent toxicity to cells in the 1st round. However, only the coated materials did not release all the drug within the first few hours and therefore showed cytotoxic effect even in rounds 2 and 3. Although in vitro release studies indicate that significant amount of the drug is no longer released at later times, experiments with cells in multiple rounds demonstrated a significant cytotoxic effect for sandwich materials compared to single one.
In accordance with the reviewer's comment, manuscript has been modified, lines 410-412.
- For biomedical applications, the stability of electrospun nanofibrous mats in physiological conditions is important and needs to be characterized. How about the stability of the PUR/PVA nanofibrous mat under physiological condition?
We agree with the Reviewer´s comment that the stability of any drug carrier plays a crucial role and it has to be considered in its development. The degradation of the polymer matrix can affect the drug release profile and, moreover, degradation products should be assessed for toxicity to the organism. In our in vitro experiments, we did not observe any disintegration or decomposition indicating limited stability. However, different behaviour can be observed under in vivo conditions where enzymatic degradation has to be considered.
In accordance with this comment, the manuscript was revised, line 365-369.
- How is TPT-loading content in the nanofibrous mat calculated? Please provide the details for calculating the TPT loading efficiency (LE), similar to that of cumulative drug release (equation 1).
We appreciate the reviewer’s comment, the formula for calculation of LE was added (Eq. 1), line 204-212.
- Please expand the acronym PUR/PVA in the title.
We agree with the Reviewer´s comment, in this regard we have expanded the acronym PUR/PVA in the title.
- What is the pH of water used for drug release study?
We thank the Reviewer for this comment; the pH of the water was 6.7 and the information was added to the manuscript.

Reviewer 2 Report
It was a manuscript about the synthesis and evaluation of a multilayer mat for the local delivery of topotecan for the treatment of retinoblastoma. Here are some comments on this study that should be considered before publication:
1- The drug release test should be checked at a temperature similar to the eye temperature.
2- Please redraw the release curve based on the percentage of drug release.
3- You need to compare and discuss the results of your research with other similar studies.
4- Some of the references are too old, please update them.
5- What is the probable method for applying the fabricated mat for in vivo evaluation?
Author Response
Reviewer 2:
Comments and Suggestions for Authors
It was a manuscript about the synthesis and evaluation of a multilayer mat for the local delivery of topotecan for the treatment of retinoblastoma. Here are some comments on this study that should be considered before publication:
1) The drug release test should be checked at a temperature similar to the eye temperature.
We are aware that it would be more appropriate to perform drug release experiments under conditions as similar as possible to the in vivo environment, i.e. in saline or blood plasma and at 37°C. Apparently, temperature will have an impact on the diffusion-controlled drug release. However, under these conditions, the stability of TPT is very low, as we presented in our previous paper (Cocarta et al, 2019). In order to somehow compare the release profiles between the different materials, we set the conditions so that the TPT degradation was as small as possible and we could determine the amount of TPT released at each time interval. In this regard, the manuscript was modified, line 330-336.
2) Please redraw the release curve based on the percentage of drug release.
We are grateful for the Reviewer´s comment, in this regard we have redrawn the release curve.
3) You need to compare and discuss the results of your research with other similar studies.
We appreciate the Reviewer's comment, comparison with other studies would certainly be beneficial, however, to the best of our knowledge there are no reports available regarding TPT nanofibrous carriers. So, we have compared our proposed nanofibrous system with other previously reported TPT delivery systems - hydrogel carriers and PCL compressed tablets introduced by Carcaboso et al. The manuscript was extended, line 421-431.
4) Some of the references are too old, please update them.
In accordance with the Reviewer´s comment, several references were updated.
5) What is the probable method for applying the fabricated mat for in vivo evaluation?
The nanofibrous carrier can be implanted, for example, perioculary as a disc under an impermeable polymeric shell (e.g. made from poly(2-ethoxyethyl methacrylate), see Cocarta 2019) onto the posterior segment of the affected eye. The manuscript has been revised, the comment regarding the future in vivo testing has been added, line 421-431.

Reviewer 3 Report
In this work, the authors presented a method of obtaining 3-layer PUR-PVP.TPT-PUR nanofibre mats by solution electrospinning. The purpose of obtaining such a material was to evaluate the possibility of using it in a controlled drug release of topocan (TPT) system in the treatment of retinoblastoma in oncological patients. The topic of the work is very interesting, it fully fits in with the aim and scope of the journal.
My Rating:
The layout of the work is consistent, all elements of a properly edited scientific manuscript are included.
The introduction contains information on the potential use of the obtained materials and the problems that occur in the treatment of retinoblastoma. Further on, the authors mentioned other alternative methods of controlled release of drugs in chemotherapy of oncological patients, and then described the purpose of using TPT in their research.
Materials and methods
In this part, the authors gave the chemicals they used and described in detail the various stages of obtaining nanofibers by electrospinning. The next section presents the research methodology.
Results and discussion
In this part of the manuscript, the authors described in detail the morphology of the obtained materials (by SEM) and then analyzed the impact of TPT release on its content in the material. The final part of this chapter is a description of in vivo and in vitro studies.
Conclusions
It contains the most important information and conclusions from the conducted research
References
The literature is well chosen.
Comments:
1. The English language needs to be corrected
2. Table 2: please unify the units in Content of TPT: theoretical (ug/disc) and actual (%)
Author Response
Reviewer 3
Comments and Suggestions for Authors
In this work, the authors presented a method of obtaining 3-layer PUR-PVP.TPT-PUR nanofibre mats by solution electrospinning. The purpose of obtaining such a material was to evaluate the possibility of using it in a controlled drug release of topocan (TPT) system in the treatment of retinoblastoma in oncological patients. The topic of the work is very interesting, it fully fits in with the aim and scope of the journal.
My Rating:
The layout of the work is consistent, all elements of a properly edited scientific manuscript are included.
The introduction contains information on the potential use of the obtained materials and the problems that occur in the treatment of retinoblastoma. Further on, the authors mentioned other alternative methods of controlled release of drugs in chemotherapy of oncological patients, and then described the purpose of using TPT in their research.
Materials and methods
In this part, the authors gave the chemicals they used and described in detail the various stages of obtaining nanofibers by electrospinning. The next section presents the research methodology.
Results and discussion
In this part of the manuscript, the authors described in detail the morphology of the obtained materials (by SEM) and then analyzed the impact of TPT release on its content in the material. The final part of this chapter is a description of in vivo and in vitro studies.
Conclusions
It contains the most important information and conclusions from the conducted research
References
The literature is well chosen.
Comments:
- The English language needs to be corrected
We appreciate the Reviewer's comment. We have carefully checked the manuscript to make language corrections, and several parts have been modified. The manuscript has been English proofread by AJE, the certificate is available.
- Table 2: please unify the units in Content of TPT: theoretical (ug/disc) and actual (%)
We are thankful for the Reviewer´s comment, the units were unified.

Round 2
Reviewer 1 Report
The authors have revised the original manuscript and addressed most of the concerns of the reviewers, other than performing the in vivo experiments. This manuscript can be accepted for publication after addressing the following points.
Comments:
1. The authors can provide experimental results demonstrating the in vitro stability of the PUR/PVA nanofibrous mat, as a function of time.
2. The sample codes in the figure captions and texts in the figures (Figures 2, 4, &5) are not consistent. Please check and correct it accordingly.
3. In addition to describing the conventional advantages of nanofibrous mat compared to the other form of drug delivery systems (hydrogel system or compressed tablets) for in vivo applications, the authors can outline distinct further in vivo experiments concerning ways of the implant administration.
Author Response
Responses to the Reviewer’s comments
The authors have revised the original manuscript and addressed most of the concerns of the reviewers, other than performing the in vivo experiments. This manuscript can be accepted for publication after addressing the following points.
Comments:
- The authors can provide experimental results demonstrating the in vitro stability of the PUR/PVA nanofibrous mat, as a function of time.
We appreciate the reviewer’s comment, the stability of PVA and PUR single layers and PUR/PVA sandwiches as a function of time was performed. In this regard, changes in the manuscript have been made. The methodology has been added on lines 167-178 and the results regarding stability have been discussed on lines 297-315.
- The sample codes in the figure captions and texts in the figures (Figures 2, 4, &5) are not consistent. Please check and correct it accordingly.
We thank the Reviewer for this comment; the captions and Figure legends have been unified.
- In addition to describing the conventional advantages of nanofibrous mat compared to the other form of drug delivery systems (hydrogel system or compressed tablets) for in vivo applications, the authors can outline distinct further in vivo experiments concerning ways of the implant administration.
We appreciate the Reviewer's comment, the changes in this regard have been made in lines 464-468.
